# Cerebral CT Perfusion in Acute Stroke: The Effect of Lowering the Tube Load and Sampling Rate on the Reproducibility of Parametric Maps

**DOI:** 10.3390/diagnostics11061121

**Published:** 2021-06-19

**Authors:** Georgios S. Ioannidis, Søren Christensen, Katerina Nikiforaki, Eleftherios Trivizakis, Kostas Perisinakis, Adam Hatzidakis, Apostolos Karantanas, Mauricio Reyes, Maarten Lansberg, Kostas Marias

**Affiliations:** 1Computational BioMedicine Laboratory (CBML), Foundation for Research and Technology—Hellas (FORTH), 70013 Heraklion, Greece; kat@ics.forth.gr (K.N.); trivizakis@ics.forth.gr (E.T.); perisynk@uoc.gr (K.P.); karantanas@med.uoc.gr (A.K.); kmarias@ics.forth.gr (K.M.); 2Department of Radiology, Medical School, University of Crete, 70013 Heraklion, Greece; 3GrayNumber Analytics, 23435 Lomma, Sweden; sorench@gmail.com; 4Department of Medical Physics, University of Crete, 70013 Heraklion, Greece; 5Department of Radiology, Aristotle Medical School of Thessaloniki, 54124 Thessaloniki, Greece; adamhatz@hotmail.com; 6Department of Medical Imaging, University Hospital, 70013 Heraklion, Greece; 7Artorg Center, University of Bern, 3008 Bern, Switzerland; mauricio.reyes@med.unibe.ch; 8Department of Neurology, Stanford University, Stanford, CA 94305, USA; lansberg@stanford.edu; 9Department of Electrical & Computer Engineering, Hellenic Mediterranean University, 71410 Heraklion, Greece

**Keywords:** Pearson’s correlation, cerebral CT perfusion, quantitative CT perfusion, lower mAs simulation

## Abstract

The aim of this study was to define lower dose parameters (tube load and temporal sampling) for CT perfusion that still preserve the diagnostic efficiency of the derived parametric maps. Ninety stroke CT examinations from four clinical sites with 1 s temporal sampling and a range of tube loads (mAs) (100–180) were studied. Realistic CT noise was retrospectively added to simulate a CT perfusion protocol, with a maximum reduction of 40% tube load (mAs) combined with increased sampling intervals (up to 3 s). Perfusion maps from the original and simulated protocols were compared by: (a) similarity using a voxel-wise Pearson’s correlation coefficient r with in-house software; (b) volumetric analysis of the infarcted and hypoperfused volumes using commercial software. Pearson’s r values varied for the different perfusion metrics from 0.1 to 0.85. The mean slope of increase and cerebral blood volume present the highest r values, remaining consistently above 0.7 for all protocol versions with 2 s sampling interval. Reduction of the sampling rate from 2 s to 1 s had only modest impacts on a TMAX volume of 0.4 mL (IQR −1–3) (*p* = 0.04) and core volume of −1.1 mL (IQR −4–0) (*p* < 0.001), indicating dose savings of 50%, with no practical loss of diagnostic accuracy. The lowest possible dose protocol was 2 s temporal sampling and a tube load of 100 mAs.

## 1. Introduction

Computed tomography perfusion (CTP) is a quantitative imaging technique for detecting hypoperfused brain regions. Following acquisition of the dynamic CTP scan, parametric maps of hemodynamic parameters are calculated and used in the diagnosis of stroke, vessel disease, and brain tumors.

There is a need to standardize patient exposure from CT examinations in order to prevent high dose accumulation, especially in cases where CT is the main diagnostic tool [1]. Moreover, due to the dynamic nature of the CTP acquisition, which involves multiple scans of the same body region, it is a widely accepted goal to find the best compromise between exposure and image quality. Based on the ALARA principle, it is important to modify the CTP acquisition protocol towards decreasing patient exposure while preserving the diagnostic value of the CTP study.

Many studies in the literature have examined the reduction of radiation exposure in CT examinations by simulating low-dose scans from higher-dose scans. These studies involve adding statistical noise (Gaussian, Poisson, or a combination of Poisson and Gaussian) either to the image intensity values or to the line integrals (raw data) [2,3,4,5,6]. Concerning CTP, the majority of published studies have incorporated either lower tube current time product simulations [7,8] or reduced image sampling rates by removing images from the temporal domain [9,10]; however, to the best of our knowledge, there has been no effort to study the combined effects of lower tube current and reduced sampling in order to address the unmet clinical need of lower-dose CTP. Besides being a two-parameter problem, by jointly considering two independent parameters towards dose reduction, we also investigate how to achieve the desired lower dose levels without exaggerated low choices in the acquisition of a single parameter that may not correspond to the actual settings used in current clinical practice. 

In this study, we investigate lower dose settings by simultaneously altering these two parameters while avoiding compromising the temporal resolution to a degree where the perfusion curve is significantly affected. To achieve this, we combine a previously tested method for realistic lower tube current simulation with reductions in the image sampling rate. To test the simulated low dose settings, this study investigates the diagnostic value of computed parametric cerebral perfusion maps in stroke patients involving lower exposure. The derived parametric maps from the original acquisition and the simulated lower-dose acquisitions are calculated using an in-house software and then are evaluated in terms of the Pearson’s correlation. In a second analysis, the effects of sampling rate reduction and lower exposure (lower tube current) are quantified in terms of the volumetric differences in the affected cerebral region between lower-dose protocols and the original protocol. 

## 2. Materials and Methods

### 2.1. Data Description

The data used in this study were a subset of the 2018 Ischemic Stroke Lesion Segmentation (ISLES) challenge dataset (www.isles-challenge.org, accessed on 17 June 2021) [11,12,13] from four clinical sites. The cohort consisted of 103 acute stroke patients who presented within eight hours of stroke onset. All patients underwent diffusion MRI within an interval of three hours after CTP examination. The image data for each patient were provided with the corresponding quantitative perfusion maps for cerebral blood flow (CBF), cerebral blood volume (CBV), mean transit time (MTT), and time to maximum enhancement (TMAX), with a matrix size of 256 × 256. The provided image data were also motion-corrected. We studied a 90-patient subset that had high frame rate acquisitions (1–1.34 s). The studied cohort was divided into four groups according to the specific acquisition protocol and scanner vendor, as shown in Table 1. Furthermore, a flowchart depicting how the patients enrolled in this study is shown in Figure 1 below.

The mean age of the cohort was 68 ± 14 years, the median baseline National Institutes of Health Stroke Scale (NIHSS) score was 16 (IQR 11–19), and the median time from stroke onset to CT was 185 min (IQR 180–238). Twenty-nine patients received intravenous thrombolysis only, 16 underwent endovascular therapy only, 14 had both therapies, and the remaining 44 had no revascularization therapy. Further information about the patient population can be found in [11].

### 2.2. Generation of Simulated Patient Image Data Corresponding to Lower-Exposure CTP Acquisition Protocols

The main goal of this study was to investigate the diagnostic value of lower-exposure protocols through the joint effect of simulating the tube current time product reduction and increasing the image sampling interval. The CTP image data for each patient as derived from the ISLES database were used to produce eight corresponding simulated image datasets, assuming 10%, 20%, 30%, and 40% reduced exposure (mAs) in combination with doubling and tripling of the temporal sampling interval. The overall workflow for the study is depicted in Figure 2.

In order to produce images corresponding to reduced-exposure acquisitions, a previously established and validated method was used [14,15] based on Equation (1), which relates image noise (SD) with the tube current time product [16]. A brief description of the method is presented below.
(1)$$\mathrm{SD}∼\frac{1}{\sqrt{\mathrm{mAs}}}$$

Assuming a patient undergoes CT perfusion examination with exposure settings $\left(E_{1}\right)$ involving a specific tube current product ($E_{1}$ mAs), the noise in a ROI in these settings will be$\;\mathrm{SD}\left(E_{1}\right)$. Let us hypothesize that we want to lower the exposure settings to ($E_{2}$ mAs).

Since the main noise source is from the X-ray detection, $\mathrm{SD}\left(E_{2}\right)$ can be calculated from the noise ratio of the two exposures $\mathrm{SD}\left(E_{2}\right)/\mathrm{SD}\left(E_{1}\right)\;\approx \;\sqrt{E_{1}}/\sqrt{E_{2}}$. The noise distributions are independent, so the standard deviation of the noise distribution to be added, $\mathrm{SD}\left(E_{3}\right)$, is found from Equation (2):(2)$$\mathrm{SD}{\left(E_{2}\right)}^{2}=\;\mathrm{SD}{\left(E_{1}\right)}^{2}+\;\mathrm{SD}{\left(E_{3}\right)}^{2},$$

The image domain noise in CT can be approximated with autocorrelated Gaussian noise, as shown by Britten et al. [14]. This involves creating an image with Gaussian noise (μ=0, σ=1) and convolving it with a kernel characteristic of the spatial autocorrelation profile in clinical CT data. Finally, the noise is rescaled to $(\mu =0,\;\sigma =\mathrm{SD}\left(E_{3}\right)$) and added to the original image data. 

The ROI for noise measurement was placed by a neuroradiologist in a homogenous region, either on white or gray matter, avoiding the stroke or ischemic site. Since the method relies on the initial noise levels, the mean value and the IQR of the aforementioned ROIs were also measured for all patient groups. Simulated image data corresponding to reduced exposure were then derived by considering four % exposure reduction levels, namely 10%, 20%, 30%, and 40%. Additionally, two reduced image sampling rates (i.e., 50% and 67% reduction) were considered by excluding one out of every two and two out of every three time points from the dynamic acquisition, respectively; thus, for each patient’s image dataset, eight corresponding simulated image datasets were derived for acquisition exposures involving lower mAs values and sampling rates. 

### 2.3. Quantification of Cerebral Perfusion

In order to compute the parametric maps from image datasets derived from the standard CTP acquisition protocol and the eight simulated acquisition protocols involving lower exposure (mAs), an in-house software was developed in Python 3.5 (www.python.org (accessed on 18 June 2021)) equipped with the PyQtGraph library for visualization purposes (www.pyqtgraph.org (accessed on 18 June 2021)). The lack of robust and freely accessible tools was the reason for implementing this technique and developing a new software tool. In addition, using commercial software from different vendors may introduce bias when compared to perfusion maps [17,18,19]. The software was based on the indicator dilution theory and the central volume principle, as firstly described by Meier and Ziegler [20]. 

Considering a bolus of CA in an artery at $t_{0}=0$ to a volume of tissue, the particles of the CA follow individual paths through the volume of interest, meaning their transit times can be modeled as a probability density function or transport function h(t). The basic equation used for the description of the concentration of the CA in the tissue $C_{t}$(*t*) and the extraction of the flow $F_{t}$ (mL mL^−1^ s^−1^) is given below:(3)$$C_{t}\left(t\right)=F_{t}AIF\left(t\right)⊛R\left(t\right),\;\left(\mathrm{HU}\right)$$
where ⊛ is the convolution operator, AIF is the arterial input function, and R(t) is the residue function denoting the amount of CA that is still present in the volume of interest at time t [21]. Mathematically, this is described as $R\left(t\right)=1-\int_{0}^{t}h\left(\tau \right)d\tau$ (unitless) [22]. The gamma variate function was chosen as the transport function h(t) to account for dispersion effects [23] based on [24]:(4)$$h\left(t\right)=\left\{ \begin{array}{ll} \frac{1}{A_{1}}{\left(t-t_{1}\right)}^{a_{1}}\;e^{\frac{-\left(t-t_{1}\right)}{\sigma_{1}}}, & \left(t\geq t_{1}\right)\\ 0, & \left(t<t_{1}\right) \end{array}\right.\;(\mathrm{unitless})$$
where $A_{1}=\sigma_{1}^{1+a_{1}}\;\Gamma \left(1+a_{1}\right)$; Γ(a) is the Gamma probability density function; $a_{1}$, $\sigma_{1}$, and $t_{1}$ are related to the mean transit time and the dispersion of h(t). The trust region reflective algorithm of the SciPy library [25] (scipy.optimize.least_squares) was used in Equation (3) to obtain the optimal values for the parameters$\;(F_{t},t_{1},\sigma_{1},a_{1}$). Finally, CBF, CBV, and MTT were calculated as: (5)$$CBF=F_{t}\;(\mathrm{mL}\;{\mathrm{mL}}^{-1}\;s^{-1})$$
(6)$$MTT\;=\;t_{1}+\sigma_{1}\left(1+a_{1}\right)\;(s)$$
(7)$$CBV=MTT*F_{t}\;(\mathrm{central}\;\mathrm{volume}\;\mathrm{principle})\;(\mathrm{mL}\;{\mathrm{mL}}^{-1})$$

Further information and technical details about the calculation of the parametric maps in Equations (5)–(7) can be found in [24]. Additionally, model-free parametric maps such as the TTP and the MSI were included in the study, since they only depend on the image intensity values. TTP represents the time taken for the perfusion curve to reach its maximum. Assuming $C_{t}\left(t\right)$ to be the perfusion curve and $t_{0}$ the last time of the baseline, MSI was calculated as: (8)$$\;MSI=\frac{1}{N}\overset{t_{N}=TMAX}{\underset{t_{1}=t_{0}}{\sum }}C_{t}\left(t_{i+1}\right)-C_{t}\left(t_{i}\right)\;\left(\mathrm{HU}/s\right)$$

In order to demonstrate the results of the implemented method, our calculated parametric maps together with these provided from the ISLES dataset, calculated using commercial software, are presented in Figure 3. 

### 2.4. Comparison of Perfusion Parametric Maps

#### 2.4.1. Correlation Analysis

The perfusion maps (CBV, CBF, MTT, MSI, TTP) for each patient produced from the standard CTP acquisition protocol (i.e., the original protocols presented in Table 1) using our developed software were considered as the gold standard (GSmaps). The perfusion maps produced using the eight simulated lower-exposure image datasets were compared to GSmaps using the Pearson’s r correlation coefficient. The Pearson’s correlation coefficient (r) and the corresponding *p*-value were calculated for each patient through voxel-by-voxel analysis for every slice of each dataset (whole cerebral volume imaged).

#### 2.4.2. Volumetric Analysis

In order to quantify the impacts of dose-reduction on the volumetric estimates used in clinical decision making, we investigated the effects of the sampling rate and exposure reduction on volumetric estimates from a commercial software package (RAPID 4.9 iSchemaView, Inc., Menlo Park, CA, USA). This software uses thresholding on the RAPID–TMAX and CBF parameter maps to estimate the volumes of infarcted and hypoperfused tissue. The software calculates the hypoperfused tissue as TMAX > 6 s and the infarcted core as CBF < 30% of the normoperfused region. We compared the lesion volumes at the 4 dose levels and 2 sampling rate reduction levels to the lesion volumes in the original scans using scatter plots and by calculating the lesion volume difference between the original scan and the volume at each dose reduction level. The paired Wilcoxon rank sum test was used to compare the original volume to the volumes at the 40% dose reduction and 67% downsampling levels.

## 3. Results

The mean and the IQR (Q3-Q1 in parentheses) for noise measured for patient groups A, B, C, and D were 7.81 (2.92), 6.09 (1.93), 3.08 (0.7), and 3.20 (0.62), respectively. In order to obtain insight into the changes of the absolute values of the perfusion parameter maps with regard to mAs reduction and temporal subsampling, we summarized their mean values, as shown in Table 2.

### 3.1. Pearson Analysis

The Pearson analysis results are summarized in Figure 4 and Figure 5. The mean Pearson’s correlation coefficient r is presented for each of the four subgroups of the studied cohort. The r is plotted as a function of the % mAs reduction for every map produced from CTP acquisitions involving temporal resolution reduced by 50% (Figure 4) and 67% (Figure 5). The *p*-values for all calculated r values were found to be <10^−3^ because of the large number of the voxels involved in the estimation of r.

### 3.2. Volumetric Analysis

The TMAX lesion volumes of the original scans had a median of 48 mL (IQR, 31–65). The core lesion volumes had a median of 8 mL (IQR, 0–21). Figure 6 shows scatter plots of the original versus dose reduction volumes for infarct core and hypoperfusion volume estimates. Similarly, Figure 7 presents the scatter plots for sampling rate reductions. Table 3 tabulates the results expressed as lesion volume differences to the original scan for each dose reduction level.

A 40% exposure reduction resulted in a median increase of 0.8 mL (IQR 0–2) for hypoperfusion volumes (*p* < 0.001) and a decrease of –0.2 mL (IQR −3–0) (*p* << 0.001). For the downsampling experiments, a 67% reduction caused a non-significant increase in TMAX volumes of 0.8 mL (IQR −2–4) (*p* = 0.1) and a decrease in core volume of −1.9 mL (IQR −5–0) (*p* < 0.001). For the downsampling experiments, a 50% reduction caused a significant increase in TMAX volumes of 0.4 mL (IQR −1–3) (*p* = 0.04) and a significant but small decrease in core volume of −1.1 mL (IQR −4–0) (*p* < 0.001).

## 4. Discussion

Stroke imaging data from four different vendors with four different CTP acquisition protocols were used to investigate whether the diagnostic value in simulated lower-exposure protocols is preserved in stroke patients based on extracted perfusion maps for the standard and the simulated lower-exposure acquisition protocols. Considering the lack of free open accessible tools for CT perfusion quantification, a dedicated software was developed. The produced parametric maps were compared with the original ones in terms of statistical correlation and volumetric analysis.

In addition, the method used for lowering patient’s exposure as indicated by Britten et al. [14] is based on the initial noise obtained by the standard acquisition protocol. For this reason, the first step in our work was to present the mean noise levels for each group by averaging noise levels per patient, a necessary step for the reproducibility of our study.

The upper limit of the mAs reduction was set to 40%, since in simulations with 50% dose reduction and one-third of the temporal resolution, the resulting perfusion maps failed to highlight the pathology, especially for group A (which had the lowest tube load among all groups). In Figure 8, it is clearly indicated that any further reduction above 40% severely affects the diagnostic value, even with visual inspection of the produced maps.

Pearson’s correlation coefficient analysis revealed that the MSI (magenta line) and the CBV (red line) parametric maps were almost unaffected by the presence of additional noise, with a constant r > 0.7 for all groups when the sampling interval was doubled (Figure 4). Furthermore, the rest of the parametric maps (CBF, MTT, TTP) presented the lowest performance in terms of correlation, with values <0.6. However, when reducing the image sampling rate to one-third (Figure 5), the correlation coefficients decreased by over 10% from those in Figure 4. 

Similar studies in the literature have shown that after increasing the time interval between successive dynamic scans to over 3 s, significant miscalculations of MTT were observed [9,10]. Taking this into account, we deduce that MTT is more sensitive to noise and this is the reason why the MTT map (black line plot in Figure 4 and Figure 5) had the lowest performance in terms of the Pearson’s coefficient. 

Since groups B, C, and D had larger tube current time products (180, 170, and 170 mAs, respectively) than group A (100 mAs), their correlation coefficients were superior to the values of group A (Figure 4 and Figure 5). On the contrary, the r values for MSI and CBV maps for group A presented a magnitude of 0.7 when the temporal resolution was reduced to half (2 s temporal resolution). Similarly, the volumetric analysis also showed a modest impact of 2 s temporal resolution with a median hypoperfusion lesion volume difference of 0.4 mL. In other words, 100 mAs is the lower limit if an r value higher than 0.7 is to be maintained (Figure 4).

In terms of the absolute values of the perfusion parameters calculated using the in-house software, all of the parameters were slightly changed by the % reduction of the tube current time product (Table 2). When the temporal resolution was reduced to half CBV and MSI were doubled while reducing temporal resolution to one-third, the aforementioned parameters tripled. MTT followed the same pattern in absolute values for both 1/2 and 1/3 temporal resolution reductions (Table 2). CBF and TTP showed very subtle changes among the different lower dose schemes. Since the variations in the calculated absolute values of CBV and MSI are significant, one should use the absolute values cautiously and focus on the relative values when comparing the affected region with the contralateral normally perfused brain hemisphere. The relative changes of CBV, MTT, and TTP values between the hypoperfused and normal parenchyma are of high clinical value, as presented in Figure 8. 

The volumetric analysis showed only small volumetric impacts on TMAX volumes from both the sampling rate reduction and exposure reduction experiments, resulting in a median that was 0.8 mL smaller in volume compared to the median lesion size of 48 mL in the population. For the core volumes, the scatter plots (Figure 6) seemed to reveal a systematic tendency toward smaller core volumes with nosier scans. Using one-third of the temporal samples resulted in core volumes with a median that was 1.9 mL smaller in compared to the median core volume in the general population of 8 mL. The reason for this more prominent relationship between more noise and lower core volumes was likely related to the noise propagation characteristics of the deconvolution algorithm. We are not aware of this finding having been reported previously. 

Various efforts towards lower patient exposure in CT imaging have been made in previous studies from different perspectives. A fraction of them have involved adding statistical noise (Gaussian, Poisson, or a combination of Poisson and Gaussian) either to the image intensity values or to the line integrals (raw data) [2,3,4,5,6]. Othman et al. [26] claimed that tube current reductions down to 72 mAs do not affect the diagnostic accuracy of CT perfusion maps. Another study presented a novel reduced-dose CT simulation technique, providing realistic low-dose images without the use of raw sinogram data [27].

Current AI-based efforts towards reducing radiation doses are limited to CT reconstruction, and have not been applied to CTP as yet [28,29,30].

Regarding CTP, the majority of published studies have incorporated either lower mAs simulations [7,8] or reduced image sampling rates by subtracting images from the temporal domain [9,10]. More specifically, Wintermark et al. investigated perfusion parameter accuracy in terms of altering the sampling intervals and the contrast volume administered to patients and compared mean and standard deviation values for the perfusion parameters in different regions. The results indicated that temporal sampling over 1 s can be used without altering the quantitative accuracy of CTP [9]. Using a similar approach, Wiesmann et al. investigated the diagnostic accuracy of CTP considering only the frequency of the sampling interval in eight patients, resulting in less than 10% of quantitative differences in measurements taken in up to one image per 3 s [10]. In contrast to these studies, we investigated the effects of dose reduction using both the correlation of perfusion parameters and by comparing the observed lesion volumes, which is of key importance in clinical practice today, across dose reduction levels.

There are factors that limit the generalizability of our study. Although the protocols used were actual clinical protocols, many sites do not sample at 1 s temporal resolution and it is likely their sampling rates will not be able to decreased any further. This would be true for most continuous spiral and toggle table protocols in use today, which have typical sampling intervals of 2–3 s. Similarly, many sites would not use as high doses as the protocols we used in our study. For sites that do use 1 s protocols and exposure levels in the range of our study, our results indicate that a change to 2 s protocols is likely to have very modest effect on the perfusion maps while halving the dose. Another limitation to the generalization of these results is the low number of patients in groups A, B, and C (N = 19, 14, and 17, respectively), since a larger patient cohort could support our methodology with greater statistical power. Despite the low number of patients in these groups, the results are consistent with the results concerning group D, with the largest number of patients (N = 40). Last, although this study was focused on stroke patients, it could be extended to other CTP examinations, such as hepatic perfusion, but this needs to be further investigated.

## 5. Conclusions

In conclusion, stroke patients could potentially benefit from lower-exposure CT perfusion acquisition protocols, since reduction of the sampling rate by 50% was found to produce perfusion parametric maps of equivalent diagnostic value compared to the initial acquisition protocols for all patient groups (groups A–D). Our study showed that high temporal resolution protocols of 1 s provided no clear benefits and sampling rates of 2 s appeared to be adequate, as long as the tube current time product was sufficient. The lowest possible dose scheme according to our results suggested an image sampling interval of 2 s and 100 mAs tube current time product.

## Figures and Tables

**Figure 1 diagnostics-11-01121-f001:**
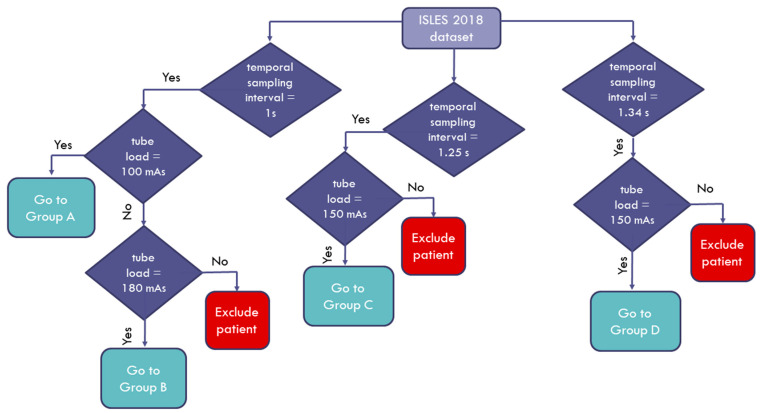
Flowchart diagram of the patient enrollment process.

**Figure 2 diagnostics-11-01121-f002:**
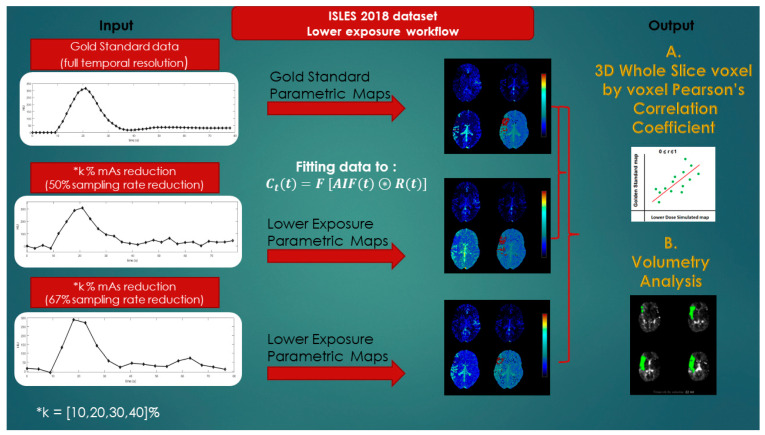
Lower-exposure workflow.

**Figure 3 diagnostics-11-01121-f003:**
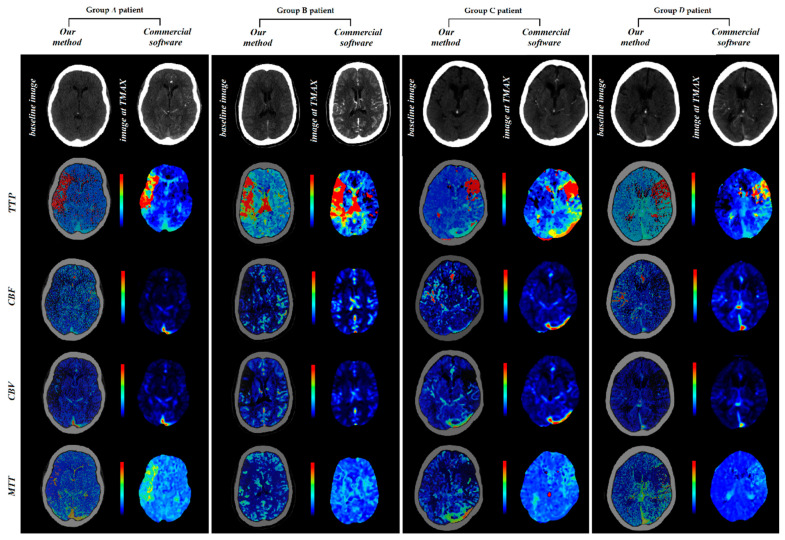
Perfusion maps calculated using in-house-developed software and commercial software.

**Figure 4 diagnostics-11-01121-f004:**
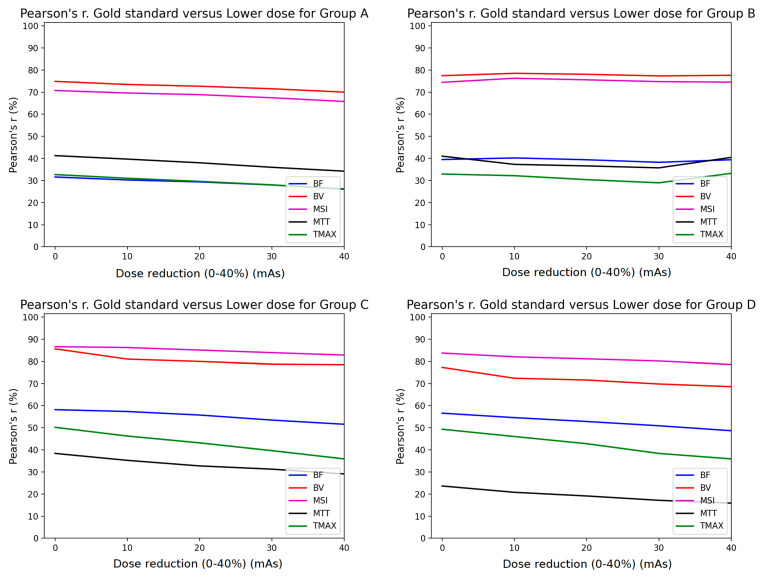
Comparison of Pearson’s correlation coefficient between GSmaps and the maps corresponding to lower mAs settings in combination with 1/2 temporal sampling for all study groups (**A**–**D**).

**Figure 5 diagnostics-11-01121-f005:**
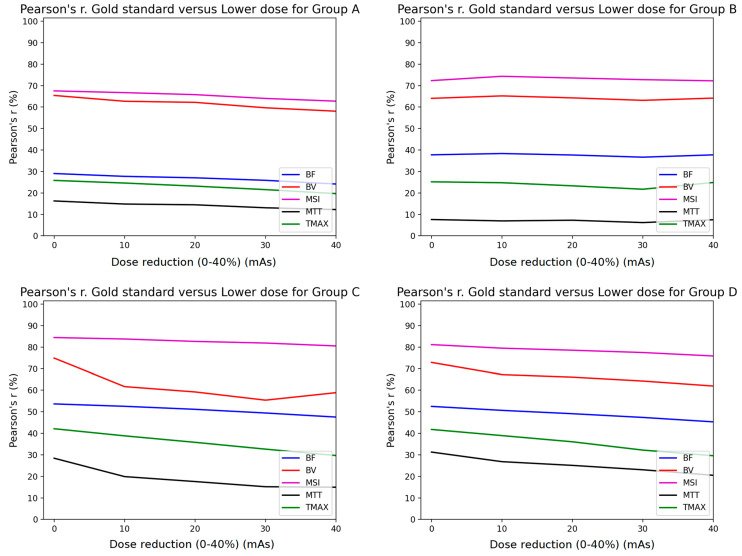
Comparison of Pearson’s correlation coefficient between GSmaps and the maps corresponding to lower mAs settings in combination to 1/3 temporal sampling for all study groups (**A**–**D**).

**Figure 6 diagnostics-11-01121-f006:**
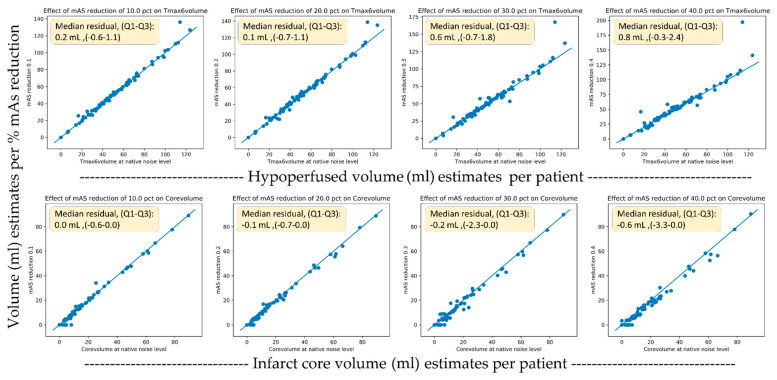
Scatter plots of the original versus dose reduction volumes for infarct core (lower row) and hypoperfusion (upper row) volume estimates.

**Figure 7 diagnostics-11-01121-f007:**
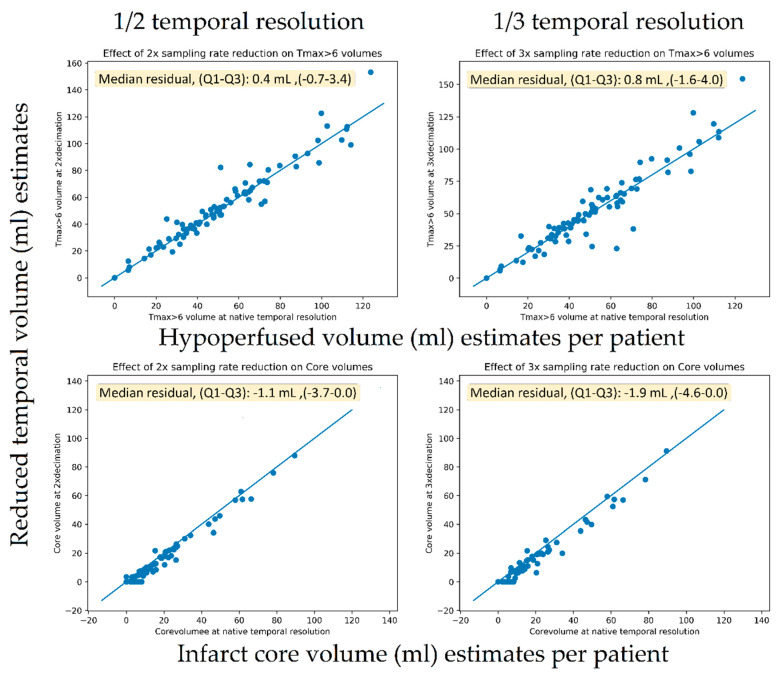
Scatter plots of the original versus temporal reduced volumes for infarct core (lower row) and hypoperfusion (upper row) volume estimates.

**Figure 8 diagnostics-11-01121-f008:**
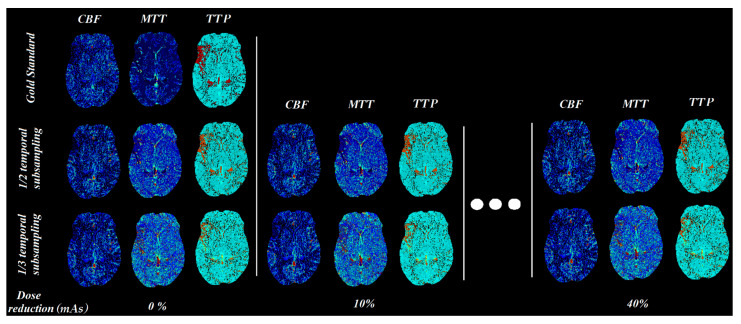
CBF, MTT, and TTP parametric maps for different mAs settings (simulated by adding noise and temporal sub-sampling).

**Table 1 diagnostics-11-01121-t001:** Patient groups and imaging parameters for each CT scanner.

	Group A	Group B	Group C	Group D
Scanner Model	General Electric: LightSpeed VCT	General Electric: LightSpeed VCT	Philips: Mx 8000 IDT16	Philips: Mx 8000 IDT16
Scan length (cm)	2–8	2–8	2.4–4.8	2.4–4.8
Slice thickness (mm)	5	5	13	13
Tube voltage (KVp)	80	80	90	90
Tube current (mA)	100	180	170	170
Rotation time (s)	1	1	0.88	0.88
Tube load (mAs)	100	180	150	150
Temporal sampling interval (s)	1	1	1.25	1.34
Dynamic Scans	50	45	35	35
Number of patients	19	14	17	40

**Table 2 diagnostics-11-01121-t002:** Mean values of parametric maps with respect to dose and temporal resolution reductions.

		1/2 Temporal Resolution				
Perfusion Parameters	Original Protocol	0% mAs Reduction	10% mAs Reduction	20% mAs Reduction	30% mAs Reduction	40% mAs Reduction
CBV (mL mL^−1^)	0.361	0.739	0.756	0.759	0.772	0.779
CBF (mL mL^−1^ s^−1^)	0.199	0.182	0.187	0.19	0.194	0.196
MTT (s)	3.138	5.351	5.332	5.278	5.254	5.209
MSI (HU/s)	0.912	1.751	1.743	1.751	1.766	1.794
TTP (s)	20.346	17.796	17.872	17.831	17.814	17.664
		**1/3 Temporal Resolution**				
CBV (mL mL^−1^)	0.361	1.22	1.323	1.336	1.379	1.431
CBF (mL mL^−1^ s^−1^)	0.199	0.199	0.204	0.205	0.209	0.211
MTT (s)	3.138	5.413	5.632	5.618	5.763	5.886
MSI (HU/s)	0.912	2.299	2.281	2.305	2.322	2.354
TTP (s)	20.346	18.908	18.979	18.94	18.92	18.796

**Table 3 diagnostics-11-01121-t003:** Hypoperfusion and core volume differences as compared to the original acquisitions for all exposure and sampling rate reduction levels.

Parameter	10% mAs Reduction	20% mAs Reduction	30% mAs Reduction	40% mAs Reduction	50% Sampling Rate Reduction	67% Sampling Rate Reduction
Hypo perfusion lesion volume difference to original scan in mL (IQR)	0.2	0.1	0.6	0.8	0.4	0.8
	(−0.6–1.1)	(−0.7–1.1)	(−0.7–1.8)	(−0.3–2.4)	(−0.7–3.4)	(−1.6–4.1)
Core lesion volume difference to original scan in mL (IQR)	0	−0.1	0.6	−0.2	−1.1	−1.9
	(−0.6–0.0)	(−0.7–0.0)	(−2.3–0.0)	(−3.3–0.0)	(−3.7–0.0)	(−4.6–0.0)

## Data Availability

The examined CT perfusion dataset titled “ISLES Challenge 2018—Ischemic Stroke Lesion Segmentation” is available online as an open-access repository via the following link: http://www.isles-challenge.org, accessed on 18 June 2021. In addition, python code for the simulations and CT perfusion quantification are available upon request to the corresponding author.

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
