# Peer review of "Cerebral CT Perfusion in Acute Stroke: The Effect of Lowering the Tube Load and Sampling Rate on the Reproducibility of Parametric Maps"

_diagnostics, 2021, doi:10.3390/diagnostics11061121_

Round 1
Reviewer 1 Report
This study shows a novel software-based way of improving CTP.
- Introduction - abbriviation CTP should be explainded when first used as it is in rest of the manuscript
- Methods- It would be of benefit to readers to see a flow chart of how patients were included and divided. Were the subdivided group homogenous ? I thin that a table presenting some clinical data including stroke severity woudl be useful. In Figure 2why there are only patients from group B and C shown ? Shoulden't you present all groups ? ALso in figure 2 last columne showing basline imaging when it comes to my opinion presents diffrent images- are there a comparison of MIP ? Please clarify figure should be selfexplanatory.
- Results - Figure 5 might be improved be enlarging subtitles, i.e. changing the entire lower line of "x" axis and giving one big subtitle. as the same for axis "y"
- Conclusions are to vague- should propose some practical implications and proposle of disseminasion of this software
Author Response
Response to Reviewer 1.
- Introduction - abbriviation CTP should be explainded when first used as it is in rest of the manuscript
- Answer: We thank the reviewer for the constructive comments. We added the CTP abbreviation in the introduction. Also, the remaining abbreviations are reported at the end of the manuscript.
- a) It would be of benefit to readers to see a flow chart of how patients were included and divided. Were the subdivided group homogenous ? I thin that a table presenting some clinical data including stroke severity woudl be useful.
- Answer: Thank you for this useful comment. Regarding the question about the homogeneity of the groups we want to clarify that all patient groups were homogeneous in terms of acquisition protocol. Following the reviewer’ suggestion we added a detailed flow chart of how patients were included and divided (page 3). This highlights the homogeneity in the subdivided groups. To reflect the changes made, we also added the following text in the data description section (page 2, last paragraph).
“The mean age of the cohort was 68 ±14 years, the median baseline National Institutes of Health Stroke Scale (NIHSS) score was 16 (IQR 11-19), median time from stroke onset to CT was 185 min (IQR 180-238). Twenty-nine patients received intravenous thrombolysis only, 16 underwent endovascular therapy only, 14 had both therapies and the remaining 44 had no revascularization therapy. Further information about patient population can be found in [11]”
- b) In Figure 2why there are only patients from group B and C shown ? Shoulden't you present all groups ? ALso in figure 2 last columne showing basline imaging when it comes to my opinion presents diffrent images- are there a comparison of MIP ?
- Answer: Thank you for this comment. We added images for representative patients for group A and D as requested. Figure’s 2 purpose is to show that our implemented perfusion quantification method produces meaningful results and there was no intention on comparing these software packages quantitatively since it is well known that different software packages produce different results [ref1]-[ref3]. Our in-house software package does not produce Maximal Intensity Projection images so these are not in the figure.
- [ref1] Kudo, K.; Sasaki, M.; Yamada, K.; Momoshima, S.; Utsunomiya, H.; Shirato, H.; Ogasawara, K. Differences in CT perfusion maps generated by different commercial software: quantitative analysis by using identical source data of acute stroke patients. Radiology 2010, 254, 200–209, doi:10.1148/radiol.254082000.
- [ref2] Zussman, B.M.; Boghosian, G.; Gorniak, R.J.; Olszewski, M.E.; Read, K.M.; Siddiqui, K.M.; Flanders, A.E. The relative effect of vendor variability in CT perfusion results: a method comparison study. Am. J. Roentgenol. 2011, 197, 468–473, doi:10.2214/AJR.10.6058.
- [ref3] Cremers, C.H.P.; Dankbaar, J.W.; Vergouwen, M.D.I.; Vos, P.C.; Bennink, E.; Rinkel, G.J.E.; Velthuis, B.K.; van der Schaaf, I.C. Different CT perfusion algorithms in the detection of delayed cerebral ischemia after aneurysmal subarachnoid hemorrhage. Neuroradiology 2015, 57, 469–474, doi:10.1007/s00234-015-1486-8.
- Results - Figure 5 might be improved be enlarging subtitles, i.e. changing the entire lower line of "x" axis and giving one big subtitle. as the same for axis "y"
- Answer: Thank you for this illustrative comment, we changed Figure 5 as you proposed. Now x and y axes are more clear for the reader.
- Conclusions are to vague- should propose some practical implications and proposle of disseminasion of this software
- Answer: Thank you for this comment. We enriched the conclusion section accordingly (page 13). In addition, the dissemination of the software was appended in the Data and materials availability section (before references section)

Reviewer 2 Report
The paper written by the following Authors: Georgios S. Ioannidis, Søren Christensen, Katerina Nikiforaki, Eleftherios Trivizakis, Kostas Perisinakis, Adam Hatzidakis, Apostolos Karantanas, Mauricio Reyes, Maarten Lansberg, Kostas Marias, entitled “Cerebral CT perfusion in acute stroke: the effect of lowering tube load and sampling rate on the reproducibility of parametric maps” presents an interesting study on the definition of lower dose parameters (tube load and temporal sampling) for CT perfusion that preserve the diagnostic efficiency of derived parametric map.
Although the paper is interesting, I have some major concerns:
Title
The title reflects the results presented here.
Abstract
The abstract is lacking informative conclusion. It should be written in more details.
Material and Methods
For all equations description and units should be included.
- There is no information about the conditions of patients acquisition. It should be included in the manuscript.
- There is 5mm and 13 mm slice thickness for acquired data. Why Authors decided to analyze so different medical data?
- There is no information about the boundary conditions for numerical simulations. It should be included in the manuscript.
Discussion
Discussion part should be improved. Authors did not refer to the existing knowledge.
Figures:
Figure 3 – 6 – the font size should be increased. Moreover, units should be included at axis.
Author Response
Response to Reviewer 2.
Abstract
- The abstract is lacking informative conclusion. It should be written in more details.
- Answer: We appreciate the suggestion of the reviewer regarding abstract’s conclusion and it has been modified accordingly.
Material and Methods
- For all equations description and units should be included.
- Answer: Thank you for this comment. We made the appropriate changes in the manuscript.
- There is no information about the conditions of patients acquisition. It should be included in the manuscript.
- Answer: Thank you for this useful comment. We added the following text in the data description section (page 2, last paragraph).
“The mean age of the cohort was 68 ±14 years, the median baseline National Institutes of Health Stroke Scale (NIHSS) score was 16 (IQR 11-19), median time from stroke onset to CT was 185 min (IQR 180-238). Twenty-nine patients received intravenous thrombolysis only, 16 underwent endovascular therapy only, 14 had both therapies and the remaining 44 had no revascularization therapy. Further information about patient population can be found in [11]”
- There is 5mm and 13 mm slice thickness for acquired data. Why Authors decided to analyze so different medical data?
- Answer: We would like to thank the reviewer for this comment since we believe that will further improve our manuscript. Indeed a small number of patients had slice thickness = 13mm. The reason for that selection of data was to test our hypothesis with a broad selection of real life data and protocols with the ultimate goal our results to be more generalizable.
- There is no information about the boundary conditions for numerical simulations. It should be included in the manuscript.
- Answer: Indeed a very useful comment. We added a relevant part (page 11, 3rd paragraph) in the discussion section.
“The upper limit of mAs reduction was set to 40%, since in simulations with 50% dose reduction and 1/3 of the temporal resolution, the resulting perfusion maps failed to highlight the pathology, especially for Group A (which had the lowest tube load among all groups). In Fig.8 it is clearly indicated that any further reduction above 40% severely affects the diagnostic value even at visual inspection of the produced maps”.
Discussion
- Discussion part should be improved. Authors did not refer to the existing knowledge.
- Answer: Indeed a very constructive suggestion. To address this comment, we added the following paragraphs in the discussion section (page ):
“Various efforts towards lower patient exposure in CT imaging have been made in the literature from different perspectives. A fraction of them are involved with adding statistical noise (Gaussian, Poisson or a combination of Poisson and Gaussian) either to the image intensity values or to the line integrals (raw data) [2–6]. Othman et al. [26] claimed that tube current reduction down to 72 mAs does not affect diagnostic accuracy of CT perfusion maps. Another work has presented a novel reduced-dose CT simulation technique providing realistic low-dose images without the use of raw sinogram data [27].
Current AI-based efforts towards reducing radiation dose are limited to CT reconstruction and have not been applied to CTP as yet [28–30].
Concerning CTP, the majority of published works incorporate either lower mAs simulations [7,8] or reduced image sampling rate by subtracting images from the temporal domain [9,10]. More specifically, Wintermark et al. investigated the perfusion parameters accuracy in terms of altering the sampling intervals and the contrast volume administered to the patients and compared mean and standard deviation of the perfusion parameters in different regions. The results indicate that temporal sampling over 1 second can be used without altering the quantitative accuracy of CTP [9]. Using a similar approach, Wiesmann et al. investigated the diagnostic accuracy of CTP considering only the frequency of the sampling interval in eight patients resulting in lower than 10% of quantitative differences in measurements taken up to one image per 3 s [10]. In contrast to these studies, we investigated the effect of dose reduction using both correlation of perfusion parameters and by comparing the observed lesion volumes, which is of key importance in clinical practice today, across dose reduction levels.”
Figures:
- Figure 3 – 6 – the font size should be increased. Moreover, units should be included at axis.
- Answer: We thank the reviewer for spotting this out. We increased the font size and included the units at axes respectively. We hope that now the graphs are more clear for the reader. Please note that after the 1st reviewer suggestion of including a flowchart of how patients got included in the study, the relevant figures are fig 4-fig 7.

Round 2
Reviewer 2 Report
I accept the present form of the manuscript.